# Performance Evaluation of Dorsal Vein Network of Hand Imaging Using Relative Total Variation-Based Regularization for Smoothing Technique in a Miniaturized Vein Imaging System: A Pilot Study

**DOI:** 10.3390/ijerph18041548

**Published:** 2021-02-06

**Authors:** Kyuseok Kim, Hyun-Woo Jeong, Youngjin Lee

**Affiliations:** 1Electro-Medical Device Research Center, Korea Electrotechnology Research Institute (KERI), Gyeonggi-do 15588, Korea; seokkyu502@gmail.com; 2Department of Biomedical Engineering, Eulji University, Seongnam 13135, Korea; 3Department of Radiological Science, Gachon University, Incheon 21936, Korea

**Keywords:** development of miniaturized vein imaging system, near-infrared image, relative total variation, adaptive smoothing technique, performance evaluation of image quality

## Abstract

Vein puncture is commonly used for blood sampling, and accurately locating the blood vessel is an important challenge in the field of diagnostic tests. Imaging systems based on near-infrared (NIR) light are widely used for accurate human vein puncture. In particular, segmentation of a region of interest using the obtained NIR image is an important field, and research for improving the image quality by removing noise and enhancing the image contrast is being widely conducted. In this paper, we propose an effective model in which the relative total variation (RTV) regularization algorithm and contrast-limited adaptive histogram equalization (CLAHE) are combined, whereby some major edge information can be better preserved. In our previous study, we developed a miniaturized NIR imaging system using light with a wavelength of 720–1100 nm. We evaluated the usefulness of the proposed algorithm by applying it to images acquired by the developed NIR imaging system. Compared with the conventional algorithm, when the proposed method was applied to the NIR image, the visual evaluation performance and quantitative evaluation performance were enhanced. In particular, when the proposed algorithm was applied, the coefficient of variation was improved by a factor of 15.77 compared with the basic image. The main advantages of our algorithm are the high noise reduction efficiency, which is beneficial for reducing the amount of undesirable information, and better contrast. In conclusion, the applicability and usefulness of the algorithm combining the RTV approach and CLAHE for NIR images were demonstrated, and the proposed model can achieve a high image quality.

## 1. Introduction

In the medical field, venipuncture is one of the critical and fundamental approaches for examination of health conditions and precise treatment monitoring of diseases, and involves blood sampling analysis or intravenous medication administration through intravenous catheterization [1]. Improper venipuncture can induce medical side effects such as swelling, bleeding, permanent vein damage, and infection, as well as erroneous examination results; hence, this procedure is performed by nurses, medical practitioners, and medical laboratory scientists [2]. Furthermore, if a patient’s vein is not clearly visible, multiple attempts are needed; thus, the patient may experience severe pain several times. It is frequently affected by the patient’s age, patient’s health condition (diabetes and degree of obesity), environmental conditions (brightness and light color), and experience of the clinical practitioners. To solve and minimize the problems induced by inappropriate venipuncture, vein viewer technologies have been designed and developed for the clear visualization of dorsal veins, which are commonly used as venipuncture sites [3,4]. Currently, most of the vein imaging technologies implement the optical property of the absorption of near-infrared (NIR) light by a human vein based on the principle of light–tissue interaction. It is well known that veins contain deoxygenated hemoglobin with a higher infrared (IR) absorption coefficient, and arteries contain oxygenated hemoglobin. These two components have optically different absorption spectrum profiles. Hence, the vein blood vessel pattern image acquired by the NIR light coming back from a biological tissue commonly appears darker than that in other regions, because the vein vessel absorbs more NIR light than the surrounding tissues [5,6]. Light in the wavelength range of the optical window of biological tissues can penetrate biological tissues at a greater depth [5,7]. However, the system performance (signal-to-noise ratio or sensitivity) can differ from the performance optimization of the two core components: the detection sensor and light source [8,9].

Many methods of identification of the veins have been investigated for the automatic insertion of a catheter into the vein, most of which are based on either the preprocessing method or the vein extraction method [10,11]. Several vein feature extraction methods have been extensively investigated, such as line tracking (e.g., repeated line tracking [12], which involves moving a dark pixel along the dark line); the maximum curvature method [13], wherein four-direction cross-sectional profiles (i.e., the horizontal, vertical, and two oblique directions) are used to calculate the local maxima of each cross-sectional profile; the wide line detector [14] using isotropic nonlinear filtering with adaptive thresholding; the filtering method involving Gabor filtering [15] and matched filters [16]; adaptive binarization [17,18], texture-oriented feature extraction techniques (e.g., scale invariant feature transform (SIFT) [19] and speeded-up robust features (SURF) [20]); the unsupervised machine learning method [21], and the classical convolutional neural network method [22]. Although some of them have proven to be highly effective for vein extraction, these methods require preprocessing to accurately extract the degraded NIR image from low-contrast and foreign objects that can be mistaken for veins.

The objective of applying preprocessing techniques before vein feature extraction is to improve the poor contrast for distinguishing veins from other features. Contrast-limited adaptive histogram equalization (CLAHE) [23] is divided into small blocks in nonuniform lighting conditions, and then histogram equalization is performed for each block. It is possible to overcome the disadvantage of sensitively responding to noise components using the simple histogram equalization method in the image domain. Additionally, high-frequency emphasis filtering [24], which uses the Butterworth high-pass filter in the frequency domain, is well-known for image enhancement and can produce outperforming results after CLAHE processing. A circular Gabor filter [25] was proposed for contrast enhancement, which is focused on the vein edge in both the spatial and frequency domains. These methods are effective for adjusting the contrast of the NIR image so that the veins can be distinguished from the other features; however, there is a limitation: substances with intensities similar to that of veins (hair, wounds, etc.) also emphasize contrast. Various techniques for vein recognition have been described in the literature [26,27,28].

In this study, we developed a cost-effective, miniaturized vein imaging system using a relative total variation (RTV) regularization [29]-based vein adaptive smoothing approach in a preprocessing framework for accurate vein pattern extraction. The system configuration is based on a handheld probe comprising a commercialized IR complementary metal–oxide semiconductor (CMOS) sensor module, two NIR light-emitting diodes (LEDs) (Thorlabs, Newton, NJ, USA) centered at a wavelength of 940 nm, and a digital light processing (DLP) projector (Texas Instrument, Dallas, TX, USA). A simple and effective RTV method achieves smoothing while preserving the main structure. We analyzed and evaluated the performance of several different noise reduction algorithms. In the following section, we briefly describe the RTV algorithm and present the experimental results.

## 2. Materials and Methods

### 2.1. Experimental Setup of Developed Vein Imaging System

Figure 1 shows the cost-effective and miniaturized vein viewer system configuration developed in this study.

Two NIR LEDs centered at a wavelength of 940 nm were employed as light sources to illuminate the target samples, such as the veins located at the dorsum of a hand and at the cubital fossa. The examined veins were positioned below the heart level of a human to make veins fill with blood as in the practical environment. The imaging target models had standard Korean skin characteristics of a bright yellow color, little hair, little fat, etc. The reflected light from the vein blood vessels and the surrounding tissues in the region of interest (ROI) was detected by the IR CMOS sensor module (SONY, Tokyo, Japan) after passing through a dichroic filter. The dichroic filter had an edge wavelength of 757 nm and a transmission efficiency of >93% in the range of 768–1100 nm; thus, it played the important role of an optical long-pass filter. The visible light coming from the DLP projector (Texas Instrument, Dallas, TX, USA) (which featured a digital micromirror device) was reflected at the dichroic filter and incident onto the imaging target area for clearly visualizing the invisible vein while the IR light was passing through. Commercial components of NIR LEDs (Thorlabs, Newton, NJ, USA) and IR CMOS sensors (SONY, Tokyo, Japan) (3280 × 2464 pixels) without IR-CUT filters were used to construct a cost-effective miniaturized vein viewer system. The center wavelength of an NIR light source was selected by a specification of the IR CMOS sensor sensitivity for the optical wavelength. The image quality-based system performance was evaluated and optimized with regard to the light angle, number of NIR LEDs, optical power, LED position, etc. An optical long-pass filter centered at 720 nm was placed in front of a detection sensor to enhance the SNR of the vein pattern image, because unnecessary visible light can be completely rejected by a long-pass filter (Thorlabs, Newton, NJ, USA). A commercialized focusing lens was inserted in front of the IR sensor after calculating the imaging range, view angle, focal length, and image display size of the DLP projector. The DLP working distance was slightly mechanically adjusted using three-dimensional (3D) micro-stages to match the DLP illumination area with the ROI size of the NIR light, because the size of the DLP illumination area and the ROI of the NIR light can change according to the respective optical paths. When the IR LED light was incident on the imaging samples, the reflected light from the ROI was detected by the IR sensor module and transferred to the main control processor (Raspberry Pi3 compute tool kit) with a 1.2-GHz quad cortex A53 and 400-MHz video core graphics processing unit (GPU). After image-processing algorithms were applied, the final image was displayed on the surface of the ROI of the imaging tissue by a DLP projector in a real-time display of approximately 60 frames/s.

To analyze the performance of the modified RTV algorithm for a system with an optical difference, we acquired the image results for two systems that differed only in the wavelength to replace the NIR LEDs, having a center wavelength of 850 nm with the NIR LEDs having a center wavelength of 940 nm. The acquired images were post-processed using the proposed algorithm. Furthermore, a color image sensor was added and positioned next to the DLP to obtain real images of the target samples. The color images could be acquired while the DLP was turned off. The ROIs of the color image sensor and IR sensors were mechanically adjusted to be identical even if they did not perfectly match from the center to the edge region.

### 2.2. Proposed Vein Recognition Technique

The total variation (TV) regularization method for preserving a large-scale edge during smoothing was introduced by Rudin et al. [30,31,32] and can be expressed as follows:(1)J(S)=argminS∑{|S−I|2+λ|∇S|}
where I represents an input image, and S represents a new image that is as similar as possible to I except for the edge component (denoted as |∇S|). Thus, the fidelity term |S−I|2 generates the smoothing edge while maintaining the main edge structure, and the regularization term |∇S| controls the degree of smoothing for S through the balancing parameter λ, which modulates the tradeoff between the fidelity and the regularization term. The regularization term is expressed as follows:(2)∑|∇S|=∑{|(∂xS)|+|(∂yS)|}
where ∂x and ∂y are partial derivatives. Here, TV regularization has an inherent limitation in that it is difficult to distinguish between the edge structure and the noise, which can negatively affect the results of Equation (1). Han et al. [33] solved this problem by applying the following equation to the regularizer:(3)∑μ(|∇S|),μ(p)=αp(1+αp)
where α is a positive parameter. This technique can achieve a better result without encountering the aforementioned problem; however, simple weighting has limitations for precisely controlling the salient edge components in the image. Li et al. [29] introduced RTV regularization, which is a simple yet effective method, based on the following equations:(4)J(S)=argminS∑{|S−I|2+λ(Dx(p)ℒx(p)+ε+Dy(p)ℒy(p)+ε)}
(5)Dx(p)=∑(ρ·|(∂xS)|), Dy(p)=∑(ρ·|(∂yS)|)
(6)ℒx(p)=|∑(ρ·(∂xS))|, ℒy(p)=|∑(ρ·(∂yS))|
(7)ρ=exp(−(xp−xq)2+(yp−yq)22h2), p,q∈ℤ
where Dx(or Dy) and Lx(or Ly) are general pixel-wise weights, which are controlled by multiplying a weighting function of the Gaussian distribution ρ. h is the spatial scale parameter of the window for determining the remaining detail and salient structure, and ε, which represents any small positive number close to zero, is used to prevent numerical instability. The RTV model does not require a prior type of texture; instead, it produces a novel map depending on the inherent windowed variation [34,35,36]. Its map has an effect on adaptive edge preservation for veins and removing the other texture in parallel.

Median filtering is a nonlinear technique that effectively removes noise while preserving contours. We set the size of the median mask (or window) to align each pixel of the degraded image (I) with the adjacent area of the averaged mask size and then slid the pixel to replace and apply the median value of the ROI to the corresponding pixel [37]. It is generally used as a filter for point noise (e.g., salt and pepper noise). Wiener filtering (i.e., minimum mean-square-error optimal linear filtering) is a low-pass frequency-domain filtering method that considers the probabilistic characteristics of the noise component [38] and is expressed by Equation (8):(8)$$S=\mu +\frac{\sigma^{2}-\eta^{2}}{\sigma^{2}}\left(I_{\Omega }-\mu \right)$$
where μ and σ represent the mean and standard variation of the local neighborhood area in the input image $I_{\Omega }$, and η represents the noise standard variation. Wiener filtering often exhibits better performance than other types of non-adaptive linear filtering. Results for the performance of the three algorithms, along with related discussions, are presented in Section 3.

CLAHE is one of the most effective contrast enhancement methods. Previously, an adaptive histogram equalization method was introduced to prevent severe distortion in some areas while equalizing the entire area with one histogram. However, there is a disadvantage: noise amplification is likely to occur when the pixel intensity is overcrowded in several dark-level histograms. To overcome this problem, a clip limit was set on the histogram, which can be transformed to be robust against noise in a region with low contrast. Therefore, the CLAHE method redistributes the pixel values until the limit value is reached before calculating the cumulative distribution function and then repeating the redistribution.

*k*-means clustering is widely used for separate clustering in the case of high-dimensional data. Each cluster has one centroid. Each object is assigned to the nearest center, and objects assigned to the same center gather, forming a cluster. The user has determined the number of clusters *k*, which is a hyperparameter. The corresponding formula is as follows:(9)$$J_{k}=\underset{c}{\mathrm{argmin}}\overset{K}{\underset{i=1}{\sum }}\underset{x_{j}\in m_{k}}{\sum }{\left(x_{j}-m_{i}\right)}^{2},\;X=\left\{m_{1}\cup m_{2}\ldots \cup m_{k}\right\},\;m_{i}\cap m_{j}=\emptyset$$(10)$$m_{k}=\underset{i\in m_{k}}{\sum }\left(x_{j}/n_{k}\right)$$
where X represents the dataset, $m_{i}$ represents the centroid of cluster $m_{k}$, and $n_{k}$ represents the number of points in $m_{k}$ [21,39]. *k*-means clustering is based on the expectation-maximization (EM) algorithm [40,41], which is a popular methodology for solving the ill-posed problem, i.e., (1) determining the location of the center of each cluster, and (2) identifying the cluster to which each object should belong.

Herein, an RTV regularization-based framework is proposed as a dorsal vein recognition algorithm. Figure 2 shows a simplified flowchart describing the application of this framework to an NIR image. First, the NIR image was obtained at 720–1100 nm near the IR system, and then enhancement images were produced by sequentially applying RTV regularization-based smoothing and CLAHE-based contrast enhancement. In this study, h and ε were set as 2.5 and $10^{-5}$, respectively. Finally, the vein image extracted using the *k*-means clustering algorithm was blended with the optical image captured at the same position as the NIR image. Here, *k* was 3, corresponding to the background, tissue area, and subcutaneous veins.

Image processing was performed using MATLAB^TM^ (MathWorks, Natick, MA, USA, R2020a), with the following computer specifications: central processing unit—Intel (Santa Clara, CA, USA), Xeon Platinum 8168 @ 2.70 GHz; random-access memory—Samsung, South Korea, 8 G × 4 DDR4 21300; GPU: NVIDIA, USA, GTX 1080 11 GB. The utility of the proposed image-processing method for practical applications was confirmed.

### 2.3. Evaluation of Recognition Accuracy

The objective of this study was to extract veins accurately, and to achieve this goal, we performed effective smoothing to compare veins with noise and foreign objects to distinguish them from grayscale and morphologically. Therefore, with a smaller amount of residue remaining after vein extraction, the smoothing method is more effective.

We evaluated the applicability of the designed framework for dorsal vein recognition with regard to the profile and coefficient of variation (COV) [42,43]. In this experiment, the results of the profile were analyzed from two perspectives. First, we checked the variation (except in the area of the vein). The fluctuations indicated that noise and foreign matter were not completely removed. The second aspect was the full width at half maximum of the vein. The elements that interfere with the extraction of veins should be smoothed to remove their main structures, while the characteristics of the vein should minimize blurring caused by smoothing. We investigated the results of the profile considering these two points. As another approach for quantitative evaluation, the COV is expressed as follows:(11)$$\mathrm{COV}\left(\%\right)=\frac{\sigma_{T}}{O_{T}}\times 100$$
where $O_{T}$ and $\sigma_{T}$ represent the mean and standard deviation, respectively, for the target ROI. In general, a smaller COV corresponds to fewer heterogeneous signals in the ROI, and the results indicated that the vein was extracted without noise and foreign objects.

## 3. Results and Discussion

In our previous study, we developed a cost-effective, miniaturized, handheld vein imaging system. The system comprises two NIR LEDs centered at 940 nm wavelength and uses the principle of obtaining an image from a CMOS detector through a dichroic filter and a long-pass filter by exposing an IR light source to the hand. In this study, a vein adaptive smoothing approach based on RTV regularization was applied to an established imaging system to obtain more accurate blood-vessel extraction images.

Figure 3 shows examples of the NIR image of 850 nm; the vein segmentation image without smoothing (①), with median filtering (②), with Wiener filtering (③), and with the proposed filtering using the RTV regularization (④); and the synthesized images of blending them with a light image. There were faint noises and foreign substances with manicured veins, compared with the other results. This indicates that the proposed method is effective for vein recognition. For a quantitative evaluation, Figure 4 shows the evaluated COV factors at specified locations *A* to *D* in the yellow-box regions. The ROIs were set as boxes *A* to *D* because these parts quantitatively determined the amount of additional noise relative to the manicured vein. A smaller amount of noise indicated more successful vein extraction. The evaluated COV factor of the vein segmentation image with the proposed filtering using RTV regularization was relatively low compared with the COV factors for image-processing methods ①to ③. The average COV factors of image-processing methods ① to ④ were 0.86, 0.60, 0.35, and 0.05, respectively. Here, the average COV factors represent the mean value of the calculated COV in each region. When the proposed algorithm was used, the quantitative average COV value was improved by a factor of approximately 15.77 compared with the case where the algorithm was not used. Additionally, the COV result obtained by applying the proposed algorithm was improved by factors of 11.03 and 6.39, respectively, compared with the result obtained by applying the median and Wiener filter to the obtained image. This result indicates that the proposed algorithm can effectively extract the vein lines while minimizing the distorted extraction in an NIR image of 850 nm.

Figure 5 shows the experimental results for the vein segmentation images (top) of the right hand for a 940-nm NIR image—none (①), the median filter (②), the Wiener filter (③), and the proposed filter (④)—and the fusion images with the vein image on the top (bottom). These results exhibited a similar tendency to those for the 850-nm NIR image and the vessel extraction results, indicating that excellent visibility can be obtained using the proposed algorithm. Figure 6 shows the intensity profiles measured along the line segments $\bar{AB}$ indicated in Figure 5 for the vein segmentation images: none (①), the median filter (②), the Wiener filter (③), and the proposed filter (④). The intensity profile obtained using the proposed algorithm exhibited a response of only two vein positions, compared with the profiles of other methods. The results confirmed the ability of the proposed algorithm to extract the vein, including the minimal noise component.

In this study, the results were analyzed focusing on the effective use and usefulness of the developed imaging system and algorithm when applied to venous blood vessels. The applicability to radial blood vessels was proved, and the results in arteries were not analyzed. However, the field in which this system can be used more effectively is percutaneous coronary intervention (PCI), which is closely related to the arterial system. In a study of Cesaro et al., the comparison of transfemoral and transradial-based methods when performing PCI provided scientific evidence that the use of the radial artery is effective in terms of bleeding risk and time required [44]. Therefore, in intervention using an artery including PCI, the imaging system is expected to show improved results.

For efficiently applying the blood-vessel image acquired using NIR light to the field of medical diagnosis, segmentation of the target blood vessel is important. In the study of Jang et al., the noise reduction efficiency was significantly improved as a result of visually evaluating the accuracy of blood-vessel extraction by applying an algorithm using the CLAHE approach based on adaptive histogram equalization to NIR images [45]. However, in their study, after the development of the NIR imaging system, a conventional algorithm was used to perform only visual evaluation (without quantitative evaluation). In the present study, the accuracy of vein extraction and the image quality were improved by simultaneously applying the RTV regularization method for smoothing and the CLAHE approach for contrast enhancement. In the future, on the basis of the quantitative results of this study, we plan to further investigate real-time localization using the NIR imaging system and improve the 3D structural analysis accuracy.

In this study, a vein viewer system based on NIR light developed by our research team was used to acquire an image of a blood vessel, and the possibility and effect were quantitatively analyzed by applying an algorithm that can achieve both a smoothing effect and contrast enhancement. However, based on the results of this study, it is considered that the improved blood vessel images acquired using light will need to be evaluated in the future with diagnostic angiography or ultrasound images that were objectively used. In addition, we plan to conduct additional studies for various clinical variables (skin colors, obesity, vein size, different regions, etc.), not just using a single blood vessel image.

## 4. Conclusions

The results of the study indicated that the proposed filtering method using RTV regularization for vein extraction is effective for improving the imaging performance of NIR images. This technique is expected to facilitate accurate and stable venipuncture through clear visualization of the invisible dorsal vein.

## Figures and Tables

**Figure 1 ijerph-18-01548-f001:**
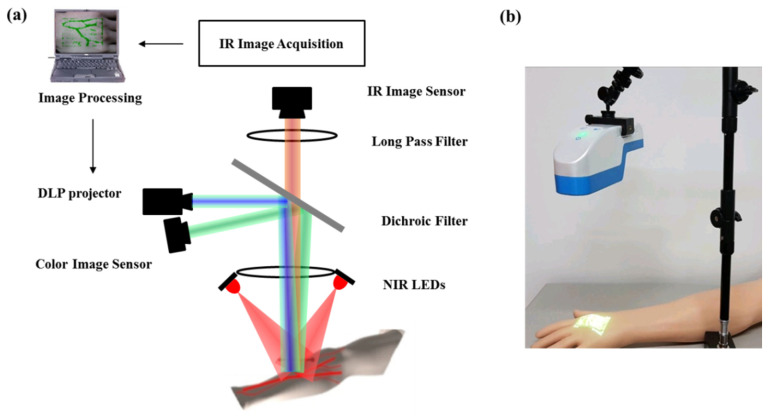
(**a**) Schematic of the developed vein viewer system configuration; (**b**) photograph of the system.

**Figure 2 ijerph-18-01548-f002:**
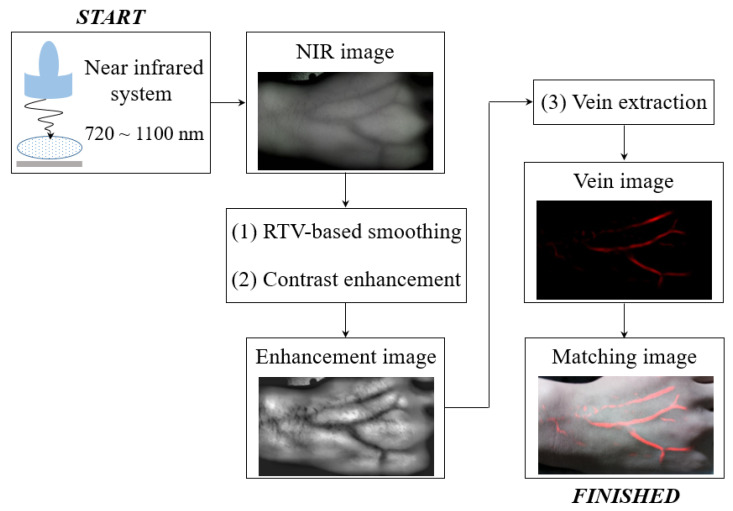
Flowchart describing the application of the proposed dorsal vein recognition framework to a near infrared (NIR) image.

**Figure 3 ijerph-18-01548-f003:**
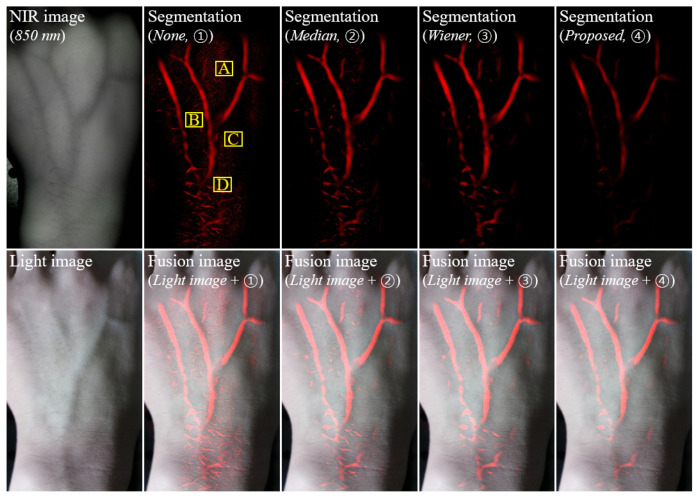
Vein segmentation images (**top**) of the left hand for an 850-nm NIR image (none ①), the median filter (②), the Wiener filter (③), and the proposed relative total variation (RTV) regularization-based smoothing algorithm (④), which were acquired using our established miniaturized vein imaging system and the synthesized images of blending them with a light image (**bottom**).

**Figure 4 ijerph-18-01548-f004:**
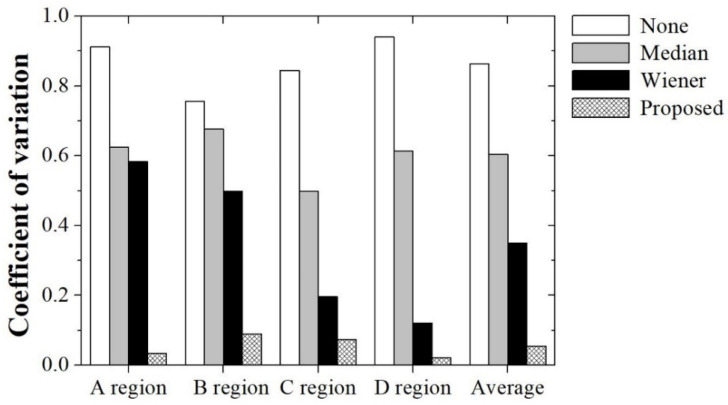
Experimental results for the coefficient of variation (COV) factors of the segmentation image under four conditions: none (①), the median filter (②), the Wiener filter (③), and the proposed filter (④). The COV was calculated for the specified regions *A* to *D* in the yellow-box regions in Figure 3. The regions of interest (ROIs) were set as boxes A to D because these parts quantitatively determined the amount of additional noise relative to the manicured vein. The average COV factors of image-processing methods ①to ④ were 0.86, 0.60, 0.35, and 0.05, respectively.

**Figure 5 ijerph-18-01548-f005:**
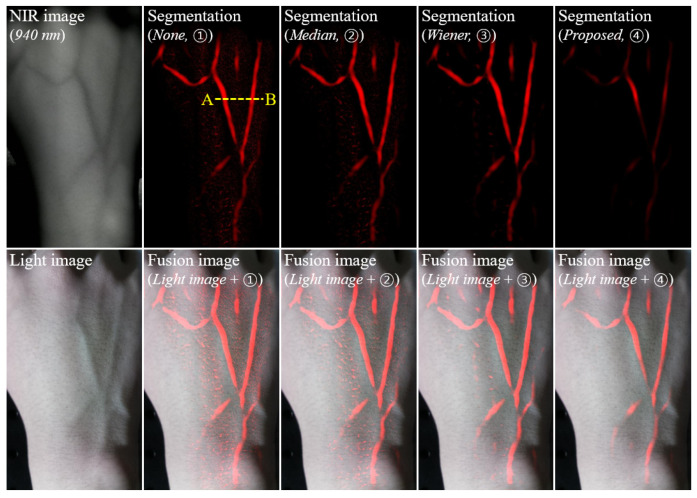
Examples of vein segmentation images (**top**) of the right hand for a 940-nm NIR image (none (①), the median filter (②), the Wiener filter (③), and the proposed filter (④) and the fusion images with the vein image on the top (**bottom**). These results exhibited a similar tendency to those for the 850-nm NIR image and the vessel extraction results, indicating that excellent visibility can be obtained using the proposed algorithm.

**Figure 6 ijerph-18-01548-f006:**
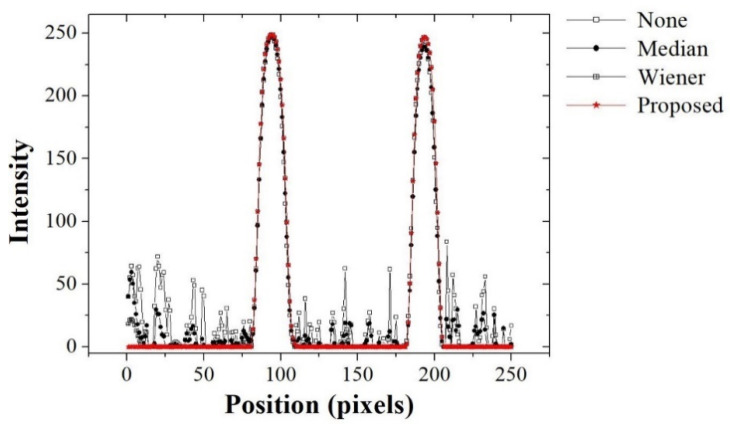
Intensity profiles (for line AB in Figure 5) for different smoothing methods applied to the acquired NIR image.

## Data Availability

Not applicable.

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
