# Peer review of "Performance Evaluation of Dorsal Vein Network of Hand Imaging Using Relative Total Variation-Based Regularization for Smoothing Technique in a Miniaturized Vein Imaging System: A Pilot Study"

_ijerph, 2021, doi:10.3390/ijerph18041548_

Round 1
Reviewer 1 Report
- Should be: „…of dorsal hand vein…” instead of “…dorsal vein..”
- Page 1. The authors wrote: “…venipuncture has recently become…” Actually this procedure by no means is new. A precursors of syringes have been described by Galen (2 century AD) and syringes and needles, albeit primitive, are in use from the 17 century. This part of the manuscript needs to be changed.
- Page 2, line 64. The authors wrote: “…Many dorsal vein recognition…”. This is confusing. The term dorsal means that the vein is located dorsally. Actually, a majority of veins that are used as vascular access points, such as cubital, jugular and femoral veins are located ventrally. I would suggest to rewrite it as: Many methods of identification of the veins….
- Material and methods. Page 3. A relative position of examined veins in relation of heart level should be described. This is very important. The veins of the hand positioned above heart level are almost empty, those below this level fill with blood. This can significantly change the findings.
- Material and methods. Page 3. There is no information regarding skin characteristics of examined hands (pale or dark, hairs, amount of fatty tissue, etc.) that potentially would make identification of a vein difficult
- Material and methods. Page 3. Line 111. The authors should explain the terms: “dorsal hand and arm” – which particular areas of the hand and the arm were assessed.
- Material and methods. Page 3. Line 120. Typo. “e a cost”
- The authors should identify and discuss weak points of their study. In particular, numerically developed images were not compared with the veins seen in the examined area with the naked eye. Neither an objective method of identification of the veins (angiography, ultrasonography, etc. )has been applied for comparison. Also, such variables as the skin color, obesity, diameter of the veins, should be discussed.
- Lastly, it should be discussed that the veins located at the dorsum of hand are rarely used for venipuncture (they are quite thin, blood flow here is minimal and puncture is painful). For blood sampling the veins of the cubital fossa are preferred, while for endovascular procedures large caliber veins, such as femoral and jugular veins are used. The veins of the dorsum of hand, primarily a distal part of the cephalic vein, are used if no other veins are available. Future research using this armamentarium should evaluate the veins of cubital fossa.
Author Response
Thank you for review and comment in this manuscript.
We have revised the paper as your suggestion and responded point by point.
Please confirm attached revised manuscript and response files.
Best regards,
Youngjin Lee

Reviewer 2 Report
In the manuscript “Performance evaluation of dorsal vein recognition using relative total variation based regularization for smoothing technique in miniaturized vein imaging system”, the authors provided an analysis of the performance of several different noise reduction algorithms a cost-effective, miniaturized vein imaging system using a relative total variation regularization-based vein adaptive smoothing approach for accurate vein pattern extraction. The topic of the manuscript is certainly interesting.
The authors found the applicability and usefulness of the algorithm achieving a high image quality with the proposed model.
The analyses were appropriate and the manuscript is well written.
No major issues were found.
However, there are some points that need further clarification:
- Please move the description (lines 250-261) to the figure caption. Likewise, for figure 5 (lines 280-290).
- Is there any data on the use of this system to improve arterial visualization? Please mention this point in the introduction or discussion, pointing out that it may be helpful in finding radial access in patients to undergo coronary angiography, since this access has many advantages over femoral access (citing the review on this topic ref. Cesaro et al. Transradial access versus transfemoral access: a comparison of outcomes and efficacy in reducing hemorrhagic events - Expert Rev Cardiovasc Ther. 2019 Jun;17(6):435-447. doi: 10.1080/14779072.2019.1627873).
- Please, you should explain each of your abbreviations the first time it appears in the main text, not only in the abstract (i.e. SIFT, SURF, ect.).
Author Response

(The authors gave the same response as above.)

Reviewer 3 Report
In the real-world clinical practice, it is sometimes challenging to correctly find appropriate vein for blood sampling or catheterization. The authors demonstrated that the their proposed filtering method using RTV regularization for vein extraction was effective for improving the imaging performance for NIR images. They concluded that the technique would be expected to facilitate accurate and stable venipuncture through clear visualization for the invisible dorsal hand vein. There are several concerns.
- The introduction section might be too long and be able to be shortened.
- In the real-world practice, dorsal hand vein is too small, unstable, and painful for venipuncture and/or catheterization. Therefore, dorsal hand vein might not be used so often for such purposes. The reviewer would like to know the reason why the authors chose dorsal hand vein instead of other veins for this investigation.
- Despite excellent visualization, it is sometimes challenging to correctly use the dorsal hand vein at a good condition. Sometimes it is occluded and dead. Clinical applicability of their technology remains uninvestigated.
Author Response
Dear reviewer,
Thank you for your useful comments and suggestions concerning our paper entitled “Performance evaluation of dorsal vein recognition using relative total variation based regularization for smoothing technique in miniaturized vein imaging system”. The revised part is marked in red in the manuscript and the response to the reviewer was explained point-by-point for each comment.
In the real-world clinical practice, it is sometimes challenging to correctly find appropriate vein for blood sampling or catheterization. The authors demonstrated that the their proposed filtering method using RTV regularization for vein extraction was effective for improving the imaging performance for NIR images. They concluded that the technique would be expected to facilitate accurate and stable venipuncture through clear visualization for the invisible dorsal hand vein. There are several concerns.
1. The introduction section might be too long and be able to be shortened.
➢ Thank you for this comment. We have reduced the content of the introduction section as you recommend.
2. In the real-world practice, dorsal hand vein is too small, unstable, and painful for venipuncture and/or catheterization. Therefore, dorsal hand vein might not be used so often for such purposes. The reviewer would like to know the reason why the authors chose dorsal hand vein instead of other veins for this investigation.
➢ We understand your concern. However, the veins located at the dorsum of hand are often used in drug injection with catheter or intravenous drip for a long-stay patient’s convenience even if those veins are quite thin and painful as you mentioned. Furthermore, it is more difficult to enhance the image quality for small veins compared over other larger veins. Hence, we wanted to study and demonstrate our developed algorithm performance for small veins at dorsum of hand as a pilot study. In a near future, we are going to study more for several skin features and veins with different size at other regions including the cubital fossa.
3. Despite excellent visualization, it is sometimes challenging to correctly use the dorsal hand vein at a good condition. Sometimes it is occluded and dead. Clinical applicability of their technology remains uninvestigated.
➢ We agree with the reviewer and in this study we demonstrated enhancement of NIR vein image quality as a pilot study. We have not examined the developed algorithm for various clinical conditions such as skin features, vein size, skin thickness, etc. Hence, in the near future, we are going to extend our study to be with various clinical vein imaging conditions as your constructive concerns.
➢ We added the following sentence in final paragraph of Results and Discussion section, “In addition, we plan to conduct additional studies for various clinical variables (skin colors, obesity, vein size, different regions, etc.), not just single blood vessel image.”
Round 2
Reviewer 3 Report
There are no further comments.
Author Response
Thank you again for reviewing this paper.